# Barriers and Challenges Affecting Quality Education (Sustainable Development Goal #4) in Sub-Saharan Africa by 2030

**Alexis Zickafoose** [1], **Olawunmi Ilesanmi** [2], **Miguel Diaz-Manrique** [2], **Anjorin E. Adeyemi** [2], **Benard Walumbe** [1], **Robert Strong** [2], **Gary Wingenbach** [2,*], **Mary T. Rodriguez** [1] and **Kim Dooley** [2]

1. Department of Agricultural Communication, Education, and Leadership, The Ohio State University, Columbus, OH 43212, USA; zickafoose.18@buckeyemail.osu.edu (A.Z.); walumbe.1@buckeyemail.osu.edu (B.W.); rodriguez.746@osu.edu (M.T.R.)
2. Department of Agricultural Leadership, Education and Communications, Texas A&M University, College Station, TX 77843, USA; wunmi.ilesanmi@ag.tamu.edu (O.I.); miguel.diaz@tamu.edu (M.D.-M.); anjy2@tamu.edu (A.E.A.); r-strong@tamu.edu (R.S.); kim.dooley@ag.tamu.edu (K.D.)
* Correspondence: wingenbach@tamu.edu

**Abstract:** Education is a fundamental human right and a crucial tool for sustainable human capacity development, which can advance the economic growth of a country. Yet for many children in sub-Saharan Africa (SSA), quality education remains out of reach. This study aimed to identify the barriers and challenges to enacting Sustainable Development Goal 4: "Ensuring Inclusive, Equitable, and Quality Education and the Promotion of Lifelong Learning Opportunities for All". Through a content analysis of the relevant literature, we identified three themes: funding constraints, access and inclusion, and teacher education. Regarding funding, issues of resource allocation, technical capacity, and accountability were identified as primary factors requiring intervention strategies to become fully realized in SSA. When exploring access and inclusion, we found that incorporating students with disabilities, gender disparities, physical barriers, and inadequate curriculum are essential factors limiting quality education in SSA. Finally, teachers' conditions and training emerged as crucial challenges to reach quality pedagogy. Some SSA countries have improved their efforts for quality education, but these barriers and challenges continue to impact education for all children. A new perspective on mitigating these impediments can address several factors responsible for the exclusion of diverse groups from accessing quality education in the SSA region.

**Keywords:** disabled students; funding; indigenous knowledge; curriculum development; sustainable development; teacher education; K-12 education; gender disparities; food systems; global south

## 1. Introduction

The United Nations General Assembly in Paris (10 December 1948) drafted and ratified the Universal Declaration of Human Rights (UDHR), which included the right to an education for all. Specifically, that "Everyone has the right to education. . .[it] shall be free, at least in the elementary stages. . .compulsory. . .Technical and professional education shall be made generally available" (Article 26 [1]). The UDHR is widely recognized as having established more than 70 human rights treaties that are permanently applied at regional and global levels today.

Since UDHR's ratification, education has evolved to meet human needs. In 1990, Education for All became a focal point because unequal access to schools prohibited millions of children from gaining basic human rights to an education [2,3]. The Sustainable Development Goals (SDGs) provide the schema for sustainable development globally. The SDGs were designed to communicate implementation results to internal and external stakeholders, and more distinctly outline and explain present and future development challenges, contrary to the original Millennium Development Goals (MDGs) [4].

Sustainable Development Goal #4, "Ensuring Inclusive, Equitable, and Quality Education and the Promotion of Lifelong Learning Opportunities for All", encourages inclusive and equitable education environments that provide quality teaching and learning, thereby promoting desire and opportunity to continue learning throughout life [5]. Quality education, comprising literacy and numeracy skills, has expanded to include vocational training, developmental skills, disability sensitivity, qualified teachers, and eliminating gender disparity, according to the United Nations [5].

Sub-Saharan Africa is falling short on achieving inclusive, equitable, quality education due to misaligned policy and funding priorities, a lack of access and inclusion, gender disparities, physical barriers, curricular constraints, and inadequate teacher training [6–15]. If these issues are not addressed, the education system in sub-Saharan African countries will continue to falter and exacerbate social and economic inequalities [16]. Causes of an exclusive, inequitable, inadequate education need to be addressed for the economic and social prosperity of individuals and their countries. Education can reshape worldviews and has the potential to address sustainability challenges [17]. This article addresses educational issues in sub-Saharan Africa by clarifying present challenges and barriers affecting quality education, with suggestions for addressing these impediments.

## 2. Materials and Methods

This study aims to better understand the barriers and challenges to quality education in sub-Saharan Africa and proffers a perspective on how to mitigate some of those challenges and barriers. Therefore, the research objective was to identify the challenges and barriers to quality education in sub-Saharan Africa.

A narrative review and content analysis was conducted to address the research objective. To ensure the trustworthiness of the research, articles were initially selected through a keyword search, followed by a critical analysis of journal articles selected [18,19]. The initial article search was conducted through the Web of Science database. The keyword search included the terms "education", "sub-Saharan Africa", "quality education", "SDGs", "barriers", "challenges", and "school" in combination to identify relevant articles. Initially, articles were restricted from 2015 to 2023 to coincide with the onset of the UN SDGs. However, upon review of the initial 20 articles, it became apparent that developing quality education in sub-Saharan Africa has been a priority since the 1990s; therefore, the timeline was expanded from 1990 to 2023, which expanded the total number of articles to 54 ($n = 54$). Articles were restricted to English-only publications.

For the content analysis, the research team inductively coded the articles based on the barriers and challenges discussed within the text [19]. Coding was conducted by five independent researchers who then came together to confirm and refine themes. These emergent codes were refined into three themes: funding constraints, access and inclusion, and teacher education. The access and inclusion theme contains sub-categories: students with disabilities, gender disparities, physical barriers, and inadequate curriculum.

## 3. Themes

### 3.1. Funding Constraints

Funding is vital to the quality of education. Its consideration is critical to achieving inclusive and equitable quality education, as stated in the fourth Sustainable Development Goal. Primary and secondary education in sub-Saharan Africa (SSA) receives less public financing than other developing regions [20]. Additionally, SSA has the highest number of children and youth out of school and is the only region where this number has increased [3]. There has been a rise of 20 million in this population from 2009 to 2021, reaching a total of 98 million people, with teenagers representing the highest proportion of the out-of-school population [3].

Successful funding interventions in SSA have been designed to tackle factors affecting infrastructure, teachers, and students' conditions [9]. These interventions range from electrification, building construction and renovation, laboratory development, the provision

of furniture, teacher training, teaching aids and pedagogy, to school management [9]. For the students, interventions consist of cash transfers, scholarships, school meals, and subsidies [9]. However, the challenges of resource allocation, resource substitution, the effectiveness of educational spending, and financial inequality are worth investigating as the gap exists beyond funding intervention and economic access to inclusive and quality education and the attainment of SDG 4.

There are three primary sources of funding in the SSA education system: local private funding from households, companies, or individual donors; external aid from international organizations; and public financing primarily from taxes, representing the primary source of education financing in SSA [21]. Considering funding, the fee-free factor necessitates discussion.

Since the early 1990s, the international community promoted a free primary education (FPE) movement [22]. The FPE basically aimed to eliminate or reduce school fees making it affordable especially for low-income families [23]. Initiatives such as the "World Declaration on Education for All" or the "United Nations Millenium Development Goals" supported the FPE movement, as well as research findings signaling the fees as barriers to accessing education [22]. Some SSA countries have attempted implementing a universal fee-free primary education system to guarantee education for all children; however, positive and negative results have been found. The fee-charging option offers the public system an additional source of income and social impact regarding parents' ownership in their children's education [24–27]; however, such fees can also be barriers for those who cannot afford them [27]. Although efforts to enhance free education have positively impacted enrollment and school completion, their repercussions on personnel and infrastructure have been detrimental [21]. The high number of students and sparse number of teachers has led to overcrowded classrooms. According to UNESCO [28], the acceptable number of students per classroom is 40 for primary and 25 for secondary school. In SSA, the average student–teacher ratio is 42 for primary school, but in countries like Chad, Ethiopia, Malawi, and the Central African Republic, averages can reach up to more than 60 students per teacher [28]. For secondary education, the acceptable ratio is estimated to be 25 students per teacher, but in the case of SSA, the average rises to 43 students per teacher. Estimates indicate that SSA needs an additional 17 million primary and secondary teachers to achieve universal education by 2030 [28,29].

Most SSA countries face limited financing capacity to increase investments in education [30]. In 2013, Africa's Gross Domestic Expenditure on Research and Development (GERD) was estimated to be USD 19.9 billion. However, several scholars have proposed that in SSA, there needs to be an almost four-fold increase in this indicator for it to reach the world average of GERD spending [30,31]. This increase is seen as a strategy to achieve educational development goals by 2030 [32]. Policymakers must commit to substantial investments in human capital, education, technology, and a skilled workforce to foster economic prosperity [19].

While SSA received approximately USD 2.1 billion as the primary beneficiary of aid for basic education in 2021, it continues to grapple with an annual financing gap exceeding USD 70 billion [33]. Although various funding interventions have been implemented, including official development assistance, multilateral aid, and private channels such as parents' spending, many African countries are still falling behind in achieving their education targets by 2030 [33]. In particular, areas such as information and communication technology, digital transformation, and school feeding programs require significant financial allocations, making it challenging for these countries to succeed [33]. Fully functioning schools with adequate infrastructure correlate positively with improved educational outcomes, yet financial constraints persist [9]. A considerable challenge arises from the strain on lower-income countries, with more than 58% grappling with rising prices and a growing debt crisis, significantly impacting education funding [34]. However, several countries in SSA do not consider the global recommendation for countries to allocate 4% to 6% of their GDP to education, urging prioritization amidst competing demands [33]. Certain countries

such as Rwanda, Zambia, and Sudan, allocate substantial portions of their budgets to debt repayment, diverting resources from education and raising concerns about resource allocation. At the same time, Ghana spent twice as much on debt servicing as it did on education in 2019, underscoring these disparities. Persistent inequality exacerbates financial access to inclusive education [35].

*3.2. Access and Inclusion*

In SSA, socio-cultural dynamics play conspicuous roles in access to education with regard to by whom, when, where, how, and what kind of education is accessed by the different members of society, especially in rural settings. In addition to funding constraints, issues of access and inclusion are at the heart of quality education disparities across SSA, which affect the enrollment and retention rates of students [13]. Access and inclusion impediments exist because of a disregard for students with disabilities, gender disparities, physical barriers, and irrelevant curriculum.

3.2.1. Students with Disabilities

Miniscule resources in SSA countries, inadequate knowledge on disabilities, and an unwillingness of governments to invest in social inclusion initiatives compound the vulnerabilities of persons living with disabilities [14]. Disabled persons in developing countries grapple with poorer health, high unemployment, lower earnings, and higher poverty, which exacerbates their vulnerabilities [14]. Children with disabilities are already disadvantaged in terms of school enrollment, educational attainment, and learning [14]. SSA countries have growing educational attainment gaps between able-bodied and children with disabilities [14]. Unable to cope with undue pressure and competition from their able-bodied counterparts, most disabled children either never enroll or drop out of school because of unaffordability, inability to access the school, social stigma, inadequate curriculum, and poorly equipped teachers [14]. Disabled children experience reduced learning outcomes; however, a dearth of inclusive educational programs and learning institutions exacerbates their vulnerabilities [14].

The concept of "education for all" within the framework of inclusive schooling refers to integrating children with special needs into the educational system alongside their typically developing peers [36,37]. While there are efforts to include students with disabilities across Africa, these efforts include providing separate classes or separate schools for these students rather than integrating them into the mainstream school system [10]. Providing the most basic assistive devices is insufficient to creating an environment where disabled children can participate in education alongside their peers [38]. This disparity highlights the shortcomings of inclusive education and the significant obstacles hindering the achievement of disability equality. In most low-income settings, there is a lack of policies and programs which mandate improving disabled persons' lives [39].

Differentiating between educational integration and educational inclusion is important as many people may use the two terms interchangeably [40]. Educational integration places disabled students in typical school environments with minimal or no curriculum modification [41]. During integration, teachers offer disabled learners extra support to perform required in-class activities so as to fit the student to the program rather than vice versa [41]. Integration does not address disabled learners' learning needs and fails to address their unique needs [42]. Educational inclusion, in contrast, restructures mainstream schooling to accommodate all children regardless of possible disability, ensuring community belonging [42]. Educational systems should offer all children equal chances for education regardless of their individual differences [40]. While the Sustainable Development Goals (SDGs) explicitly include providing equal access to education and vocational training for vulnerable individuals, including those with disabilities, the effective implementation of teaching strategies for children with diverse abilities remains a challenging endeavor in SSA countries [14]. The incorporation of education into a comprehensive human rights agenda condemns any type of segregation [42].

Current research indicates that globally, there is an increased commitment to inclusivity and equity in development initiatives through the adoption of the SDGs and the "Leave no one behind" agenda [14,38,39,43]. This momentum ensures that marginalized populations, mainly disabled persons, are actively included and considered in mainstream development endeavors [14,38,39]. Seemingly, education in SSA can equalize opportunities; however, it does not address underlying educational equity. Education for all is foundational for attaining educational equity. There is nuance in balancing setting and maintaining educational standards, while still reducing barriers, and increasing inclusion [37]. Educational systems must provide equal opportunities for all children, regardless of their underlying differences, since education is the glue binding societies [40].

Many SSA governments delegate the responsibility of enhancing the well-being of individuals with disabilities to faith-based organizations which offer specific and localized education, livelihood, and healthcare services [14]. In the short run, such humanitarian assistance seems feasible. However, eventually, such assistance contributes to compliance, and prevents governments from providing education to people living with disabilities. Moreover, the abdication of government responsibilities to faith-based organizations exacerbates the inequality gap since donors may not stay involved, leaving vulnerable regions and learners with unstable learning environments that constantly need resuscitation. Overall, governments evading responsibility is detrimental to educational sustainability to marginalized learners and regions.

Developing inclusive and equal learning environments for children with disabilities remains an elusive venture in SAA countries whose education systems grapple with innumerable challenges. Even though 166 nations endorsed the United Nations Convention on the Rights of Persons with Disabilities, execution is inconsistent among member countries [14]. Providing equal opportunities for disabled children is challenging, but inclusive education can benefit all children [14,44]. However, barriers affecting disabled students intersect all students' experiences.

### 3.2.2. Gender Disparities

Access and inclusion barriers affect girls more than boys. Gender disparity between boys and girls widens more rapidly following primary education, negatively affecting enrollment and retention rates, which allows disparities to continue into adulthood literacy rates [13,15]. These factors include curricular and extracurricular opportunities, differences in treatment by individual teachers based on sociocultural beliefs, differences in rules and regulations, assigning science, technology, engineering, and math courses to boys rather than girls, parents preferring to educate male children over female children, and administrative practices, which could all impact the quality of educational experience enjoyed based on gender [13,15,45].

Moreover, there is a significant negative correlation between cultural practices such as female genital mutilation (FGM), early marriage, and teenage pregnancy, with quality education [46–48]. Globally, over 200 million girls and women have been subjected to FGM, and 70 million girls aged 0–14 years face the risk of undergoing the same trauma each year [49]. FGM, which is still practiced in many SSA countries, has a cultural connotation of ensuring a woman's propriety and cleanliness and is considered a rite of passage [47]. FGM can decrease girls' school performance and increase their absenteeism and drop-out rates [50]. Sociocultural norms may have an impact on girls' ability to access education.

### 3.2.3. Physical Barriers

Physical access barriers also disproportionately affect girls. Physical access issues for women include inadequate sanitary facilities needed for privacy due to menstrual stigmatization and discrimination [15,45]. In addition, some schools across sub-Saharan Africa are nearly inaccessible because some students lack transportation to schools which are more than 40 min by foot [51,52]. The distance between schools and communities frequently discourages girls from enrolling in far-distanced schools since they may not be

able to complete house chores [15,45]. This is not necessarily mitigated through virtual classrooms or online options as digital technology is not accessible for many students [53]. Most rural African areas also lack access to electricity, hindering digital transformation and limiting access to computers, projectors, and interactive boards, affecting the digital literacy rates of young people in these areas [9]. This lack of access and inclusion affects disabled students more, especially regarding transportation and digital technology access and utilization.

Situations such as war, assault, and robbery affecting students on their way to school, the long distances they must travel between their homes and their school, and the hygiene of school canteens are examples of critical factors in achieving access and quality education offered to students [52,54–56]. For example, in sub-Saharan Africa, low rates of male secondary education are associated with a high risk of civil conflicts [57]. Civil security, transportation infrastructure, or public health issues are vital, affect the daily lives of students and teachers, and require financial resources for their support [58].

### 3.2.4. Inadequate Curriculum

SSA countries' curriculum contains gaps in knowledge that are socially and culturally relevant to their students. There is a scarcity of user-friendly methods that align with the appropriate curriculum [36,38,39]. Deficits in language and Indigenous knowledge may prevent students from engaging in the curriculum or seeing the benefit of participating in the educational system. Relying solely on monolingual pedagogy falls short in delivering quality education for many African children, particularly when the instructional language is secondary or foreign, impeding cognitive and academic development [11].

Translanguaging involves engaging learners in their home language, recognizing diversity, and removing barriers that may hinder participation and achievement [11]. Alderman et al. [59] notes that children learning in a language different from their mother tongue are 30% less likely to attain minimum reading proficiency by the end of primary school. Conversely, a lack of proficiency in English leads to low student participation, reduced confidence, disinterest, ineffective content delivery, and a restrictive atmosphere [11]. Despite challenges such as time constraints, unfriendly assessments, potential threats to content comprehension, numerical categorization, and the facilitation of labeling and ethnic divides among students, translanguaging brings forth benefits like enhanced content grasp, strengthened student-centeredness, improved cross-referencing, versatility, and a relaxed atmosphere [11]. The advantages of translanguaging should be considered to facilitate sustainable and quality education accessible to all learners due to the plethora of languages present in SSA.

Sub-Saharan African education systems are also missing a vital local component: Indigenous knowledge. Indigenous knowledge encompasses the practices, histories, identities, and way of life unique to a cultural group [60]. SSA education systems are often devoid of Indigenous knowledge systems [61]. The education systems present in much of Africa have been designed by European colonial powers and have endured long after individual countries' independence [61,62]. Since education inculcates people with the values, culture, and way of life of a society, a system cannot be devoid of the beliefs and values of those who created it. Following independence, African classrooms also had more foreign than native teachers, which led to the curriculum reflecting teachers' context rather than students' context [61]. Integrating Indigenous knowledge into curricula could lead to better sustainable development. By including Indigenous knowledge in the curriculum, students can be better prepared to thrive in their environment because of reintegrated local expertise and the preservation of traditional knowledge [61]. While the call for the integration of Indigenous knowledge is resounding at the university level [63,64], few efforts exist for primary and secondary education. Integrating Indigenous knowledge in the formal education curriculum could make the curriculum more relevant and better prepare students to address local problems [60].

*3.3. Teacher Education*

Teacher education has been impacted by various factors, including the duration of training, the level of ongoing support, the extent of engagement and participation, the inclusion of practical experiences, the integration of reflective elements in courses, the level of contextual adaptability, and the degree of collaboration among teachers [65]. Specifically, there are cases of teacher absenteeism, a lack of basic experience, and a lack of training and resources to equip teachers with high-quality, learner-centered pedagogy. These limit opportunities for both teachers and students in most SSA countries.

In the case of Higher Education in Africa (HE), aligning concerted efforts is crucial to achieving the purposes set in the fourth SDG [6]. Moreover, it is reported that critical efforts from governments and international agencies further undermine the attainment of the SDGs [7]. These efforts may not be removed from the inability of the HE systems in Africa to match increasing efforts in funding, enrollment, teacher training, and pedagogy with appropriate physical infrastructure, quality-based training, and the capacity building of relevant professionals crucial to achieving the SDGs [6].

In a region where resources are scarce, competent and trained staff should support the ability to prioritize resource allocation. In this regard, misplaced priorities and a need for more ability to manage the resources represent additional factors impacting education quality [66]. According to UNICEF [67], in most SSA countries, almost half of the public educational financial resources are dedicated to the most privileged students, while less privileged children do not attend the educational system or withdraw from it. To prevent unattendance and withdrawal, funding allocation in SSA should be guided by a vision of progressive universalism, targeting the most disadvantaged and vulnerable populations as the first recipients of resource allocation [68].

Research consistently emphasizes the substantial impact of investments in teachers and teaching methods on student performance [9]. However, studies indicate that the quality of teachers, including their subject knowledge and didactic skills, plays a crucial role in influencing student outcomes, especially when teachers lack competency in utilizing teaching and learning aids for effective curriculum delivery [69,70]. Despite increased enrollment rates in SSA over the past two decades, standardized testing reveals minimal improvements in student learning [70].

Notably, significant learning gaps persist among secondary school students in SSA, often attributed to a substantial deficit in teacher quality. Factors such as low teacher pay, limited access to technology (e.g., laptops), and challenging working environments, as observed in Rwanda and Nigeria, contribute to the existing disparities [12,71,72].

Significantly, the foundational prerequisite for lifelong student learning and teacher training on environmental, economic, and social sustainability concepts must be protected at the primary and secondary education level [6]. Conversely, the demand for the institutionalization and internationalization of collaborative knowledge generation and utilization among scholars, researchers, and practitioners is reasonable, but investment into research and development poses a challenge.

## 4. Discussion

The challenges and barriers to quality education in SSA impact a wide breadth of individuals. While SSA countries grapple with multiple barriers and challenges, solutions are not universal. Each SSA country has its own context and nuance; however, the barriers and challenges are similar across contexts. SSA governments lack agency, which hinders the pursuit of a unified agenda that prioritizes their citizens' developmental needs [73]. Funding constraints, competing priorities, a lack of access and inclusion for differently abled students and girls, physical barriers, inadequate curriculum, and poor teacher education all contribute to a stymied education system in SSA. Mitigating these issues can increase quality education across SSA, which can accelerate the transformation to enhanced sustainable food systems, agriculture, and individual diets through better informed decision-makers [74].

Quality education can support women and those with disabilities, which will promote other SDGs and encourage sustainable development [5,75]. Education provides numerous socio-economic benefits, such as reduced mortality rates [76], increased research and technology production [77], knowledge transfer efficiency [78], higher income [79], and the long-term economic growth of countries [80]. These benefits may have a profound impact on SSA, the region with the lowest levels of schooling, but where investments in education have the highest private and social returns [81]. The barriers and challenges to quality education in SSA are highly interconnected, and mitigating or eliminating them may take a variety of pointed interventions [82,83].

### 4.1. Addressing Funding Constraints

Funding is a determining factor for many other components of an education system. Elements such as access, teacher salaries, and educational infrastructure are permeated by the flow of financial resources. Recent data indicate the depth and amounts of limited resources that are hindering quality education for stakeholders [23]. Increased investments in education can produce increased societal development and economic growth, while societal development and economic growth can allow more investment in education.

The effective management of financial resources is just as crucial as obtaining those resources. Training the staff responsible for administrative duties is essential for the efficient allocation of the budget in the education sector [66]. Proper budget planning, implementation, reporting, and tracking ensure transparency in resource expenditure and coherence in executing programs and policies [52,66,69,84,85]. Uganda and Senegal have had successful experiences that could be considered in other countries. Uganda developed the Education Funding Agency Group (EFAG), while Senegal has the National Education Account (NEA); these organizations serve as centralized bodies of accountability, bringing together resources coming from donors in the case of the EFAG, or from private donors and the government in the case of NEA, offering support to monitoring, funding targeting, and decision-making processes involving financial resources for the education system [69]. Administration groups that emulate those in Senegal and Uganda could provide accountability and increase available resources for education in other SSA countries. These initiatives will be more critical when external aid decreases and the need for funding per student expenditure increases [3].

The education system in sub-Saharan African countries must also be seen as just one component of a much larger system and, therefore, one that affects and is affected by other factors. Since many SSA countries devote a substantial portion of their budgets to debt repayment [33], efforts should be made to decrease debt burden through diplomatic renegotiation. These efforts could allow more of SSA countries' GDP to be diverted to education. Therefore, it is not just a matter of advocating for resources for the education system but also of understanding its interdependence with other social links. Since conflict situations exacerbate access to schools for all children, increased national security and normative values could also strengthen educational funding and programs [57,86]. Increased efforts to stymie corruption such as building national and municipal capacity to investigate infraction and enforce the rule of law could also support efforts to ensure proper financial management. Critical to the attainment of this paradigm shift is the urgent need for educational policy reforms that are pragmatic, proactive, and that can break the policy syndrome that has crippled the possibility of quality education for all over the years [87].

### 4.2. Increasing Access and Inclusion

The issues regarding access and inclusion cross ability, gender, physical, and curricular divides. Interventions addressing inclusion issues may crosscut a variety of barriers and challenges. Across SSA, limited access to education by children with disabilities is persistent [15,41,42]. Limited access to schooling exacerbates the inequality gap between the able-bodied and children with disabilities, thus denying the latter the transformative power of education [40,42]. Stigma, a lack of awareness, and insufficient infrastructure

contribute to a cycle of exclusion, thus denying children an education and a chance to break free from the shackles of poverty and dependency [14,37,40].

Initiatives which aim to increase educational access for learners with disabilities that extend beyond integrating children with disabilities into existing structures are needed across SSA [36,37]. African Ubuntu values such as inclusiveness, fairness, and dignity align with human rights principles and are integral in inclusive education [10]. Increasing the prevalence of Ubuntu values within primary and secondary education in SSA could help invigorate the desire and need to include diverse learners. True inclusion involves creating environments that cater to diverse learners' needs, including investing in specialized training for teachers, providing accessible infrastructure, and fostering a culture of acceptance within schools [40,42]. A vital step towards change is raising awareness about the rights and capabilities of children with disabilities as well as revitalizing Ubuntu values in the African context, which could advance specific policies and practices for SSA education [10]. Advocacy can be pivotal in challenging stereotypes and dismantling inclusivity barriers. Initiatives that value and capitalize on student strengths and self-reliance while increasing teacher capacity and the willingness to include students with disabilities in the classroom could lead to greater inclusion. Breaking the stigma of disabilities through teacher education and preparation could lead to a reinvigorated education system that is able to support students with disabilities [39,41]. Governments, NGOs, and international organizations should forge partnerships to tackle this issue collectively. A collaborative effort could pool resources, share best practices, and develop sustainable strategies that address the unique challenges faced by each community [14]. Closing prevalent gender gaps will also require a concerted effort from all stakeholders in SSA. In countries where gender disparities in access to education have been overcome, the first step has often been to acknowledge and affirm the fundamentality of education for all citizens as a right, regardless of gender or sociocultural beliefs, and that acting contrariwise is deemed a violation of that right [15]. This cannot be overemphasized, bearing in mind that so much can be achieved and sustained when educational policies and actions are not just carefully planned and executed, but, more importantly, are enshrined within the legal or constitutional framework of each country as a right. This creates a sense of responsibility and accountability for the parents, the communities, and the government to not only uphold the law, but also to enforce it whenever there is a partial or total violation.

The huge economic opportunities being lost from knowledge and skills gaps in educational disparities by gender should also be of grave concern to leaders in SSA and serve as a call to action. If these gender gaps are not closed through equal access to quality education, especially in SSA, the impact will be felt in terms of unequal human capital development ratios, higher rates of underweight children, and soaring rates of child mortality [88–91]. Empirical evidence suggests that higher levels of female education are positively correlated to higher children's education and lower dependency ratios, which in turn leads to higher income [92,93]. Therefore, leaders in SSA should push for more comprehensive education policies and programs that leave no girl or woman behind.

Comprehensive policies could include initiatives to make school more accessible and attuned to girls' and young women's needs [13,15,45]. Developing nonformal schools with more flexibility that rely on community involvement could transition rural populations into the formal school system. Childcare schemes at local schools could encourage women and girls to attain their education post marriage and childbirth. Building private toilet facilities for girls, including gender-sensitive curricula, and hiring more female teachers could create safer learning environments for girls [13,15,45]. Stipend programs for girls attending school could increase retention and attainment. SSA countries could target gender disparities through a program similar to Bangladesh's Female Secondary School Stipend Program (FSSSP) [94]. The FSSSP was introduced by the Bangladeshi government in 1994 to make secondary education free for rural girls and provide a cash stipend for them [94,95]. Prior to FSSSP initiation, 75% of girls enrolled in primary school but only 14% enrolled in secondary school, compared to 85% and 25%, respectively, for boys, with girls experiencing

a 60% dropout rate [94]. To receive the stipend, girls needed to have a minimum of 75% attendance rate, at least a 45% on annual exams, and remain unmarried [94,95]. Since FSSSP implementation, participant girls are more likely to complete secondary school or beyond, get married and have children later, and work in the formal sector [94]. A program based on FSSSP may be applicable to SSA contexts to encourage girls' education by adding value to it and relieving parental financial burdens.

Problems of physical access to education include lengthy walking distances to school, inadequate sanitary facilities, conflict zones, and the lack of electricity and digital technology to combat physical distance. One way to combat physical distances could be using radios to deliver school content. Radios are widespread across SSA and have been used effectively to disseminate information to both urban and rural populations [96,97]. Similarly, Australia effectively used two-way radios to deliver education to remote children, called the School of the Air, until internet accessibility became widespread [98]. A version of the School of the Air may be applicable in some SSA countries since reliable electricity and internet access may not be available throughout most countries, hindering virtual options.

Inadequate curriculum may also be mitigated through culturally relevant curriculum. Initially, ensuring that the curriculum is delivered in the students' home language could increase student retention and the understanding of the content [11]. The content itself may also be redesigned. While the curriculum itself will look different across SSA, the process of designing it may look similar. One way to begin this process would be to deconstruct the current national curriculum based upon ideological critique and then reconstruct it through disrupting normativity and cultivating agency to begin the process of decolonization [61,99]. These tools of deconstruction and reconstruction can help national leaders and policymakers to identify and redress hegemonic norms and core beliefs to elevate individual and collective agency [99]. Efforts can also be made to increase Indigenous knowledge within the curriculum. Indigenous knowledge can be added through encouraging local elders to infuse curricula with experiential learning, traditional values, and cultural traditions. These efforts can shift curricula to teaching through culture rather than about culture, increasing the cultural relevance of curricula.

An education system that serves students regardless of ability, gender, income, or class while teaching a culturally relevant curriculum by properly trained teachers could allow students and teachers to reach their full potential through sustainable development [100]. Educating key actors and stakeholders about sustainability and incorporating sustainable development into school systems would have an important impact on SDGs [101]. A primary concept to accomplish sustainable development globally is referred to as ESD or Education for Sustainable Development [100]. ESD seeks to empower people of all ages to create a sustainable future through locally relevant and culturally appropriate learning [102]. Global institutions and science educators are seeking methods to improve students' experiences and the transformational education of SDGs [103].

Addressing issues of access and inclusion could increase the sustainability of education systems across SSA by creating highly informed decision-makers. Issues of physical access prevent rural populations from participating in the education system, which can be exacerbated when children have disabilities. Breaking barriers that enable discrimination due to an individual's ability and gender is a moral imperative and investment in social and economic prosperity for the entire region. The journey towards inclusivity begins with recognizing individual potential and building a society that values and embraces diversity. Only then can we claim that education is a right rather than a privilege for all.

*4.3. Addressing Teacher Education*

It is evident that the compromise in teacher quality poses a substantial drawback amid the pursuit of free education and increased enrollment in early childhood, primary, and secondary education, a trajectory observed in many developing nations, especially in SSA. Understandably, the parallel surge in teacher recruitment with rising enrollment is targeted to address student–teacher ratios [22]. However, this expansion must deliberately

seek to mitigate diminishing teacher quality standards, impacting teaching quality, student learning outcomes, and the negative economic repercussions seen across several African countries [104]. From a teacher's perspective, intentional actions to drive teaching quality across its varying dimensions, which include the teachers' ability for cognitive activation, classroom management, and teacher support, is important [105]. Importantly, teachers must be able to create a rounded environment to offer equitable, quality learning outcomes for all students [106].

Additionally, delivering a quality and valuable learning experience is premised on all students acquiring basic fundamental skills, which is contingent upon whether the teacher knows the content to be taught. In Africa, teaching is often perceived as unattractive because of low remuneration, but this perception can be reversed through intentional policies targeted at teacher training and professional development based on performance and accountability principles [107]. These targeted interventions can be represented in a variety of models. They cut across the use of customized lesson plans and training for teachers, the incorporation of new materials for students, and the use of mother tongue language instruction [105]. Policies to align the teacher's ability and practice with student learning levels are crucial to avoid mismatches between student and teacher learning styles, as theorized by Grow [108,109]. Advocacy for holistic teacher education and training is essential, encompassing both remunerative and non-remunerative programs, such as pay-for-performance initiatives and guided coaching, respectively [16]. Further findings from Evans and Mendez Acosta [16] reiterate that policies to drive teacher accountability and performance feedback can produce literacy gains and improved educational outcomes in students, especially those from a low socio-economic background. Studies in Tanzania, Kenya, Rwanda, and Ghana support this effort with evidence on improved test scores and literacy gains.

Given that developing countries have faced similar woes of declining quality education attributable to teacher quality, our perspectives align with empirical findings in India that call for multiple stakeholder engagement rather than simply advocating for teacher education or training [110]. Teachers' trajectories are limited without targeted interventions. They need ample support from the government to provide resources that can amplify their abilities by having access to top-down support and monitoring systems. Equally reinforcing is the need for teachers to cultivate the science of delivering an age-appropriate, learning-level-specific, and skills-specific learning environment for all students. Teachers play a pivotal role in either facilitating or hindering students' learning capabilities [108,109]. Therefore, efforts to enhance effectiveness should commence with teacher awareness and reflection, making teachers active participants in the design and implementation of the training activity [109]. These perspectives ring true especially for policymakers, practitioners, researchers, government stakeholders, and teachers to commit to providing effective teachers for quality education. The World Bank reported

> . . .efforts to foster effectiveness would be more productive if a teacher's awareness and reflection was used as the starting point of training efforts. Teachers should be the originators of and active participants in the design and implementation of teacher training activities. [111] (p. 157)

Soares et al. [112] also emphasized the alignment between the African Union's 2063 agenda, outlined in "The Africa we want: A Common Strategic Framework for Inclusive Growth and Sustainable Development", and the objectives and aspirations of the SDGs. Hence, recognizing the significance of teachers as fundamental pillars for the capacity and sustainability of national goals becomes imperative [113]. This importance is underscored by the challenge of deviating from the path toward achieving Sustainable Development Goals, which pales in comparison to the greater concern of students being unable to grasp these goals from their knowledge custodians—the teachers. The alignment of teacher education, ability, training, and practice using targeted policies to create age-appropriate, learning-level-specific and skills-specific environments for all students with effective motivation is pivotal to delivering quality education and promoting sustainability in SSA.

### 4.4. Limitations and Implications

The limitations of this study lie within the limits of literature reviews. As this narrative review utilized only one database, Web of Science, there is the possibility that some relevant articles could be found in other databases. In addition, during the initial search, there were some articles that may have been relevant but were not in English and therefore not able to be included in the content analysis. The lack of the inclusion of these potentially relevant articles limits the breadth of information in this narrative review. The research is also limited due to the availability, scope, and biases present in the previously published sources used to formulate this review. Therefore, future research should aim to conduct empirical studies on the quality of SSA education to validate and enhance the findings of this review. Future research should also delve deeper into the identified themes to target country-specific challenges and test potential interventions to combat those challenges.

### 5. Conclusions

Inadequate education can prevent individuals from realizing their full economic and social potential, leading to community and country stagnation. Our work supports [114]'s findings, in that achieving SDG #4, i.e., quality education, in SSA will remain farfetched unless fundamental change occurs. Quality education depends on creating an inclusive, equitable education system that incorporates socially sustainable practices. The impediments to achieving inclusive, equitable, and quality education and the promotion of lifelong learning opportunities for all in sub-Saharan Africa cross-cut economic and socio-cultural bounds. Key barriers and challenges include funding constraints, a lack of access and inclusion, and inadequate teacher education. Further, our research builds upon [114] by providing practical solutions to these interconnected and interdisciplinary issues, which depend on synergistic collaborations among Global North institutions and all Global South actors and stakeholders to improve the education systems across sub-Saharan Africa. Concerted efforts to target the key barriers and challenges may increase sustainable development across the continent that encourages the fulfillment of SDG 4, which will further impact all SDGs for individuals and communities.

**Author Contributions:** Conceptualization, G.W., A.Z., O.I., M.D.-M., A.E.A. and B.W.; methodology, A.E.A. and A.Z.; validation, G.W., K.D., M.T.R. and R.S.; formal analysis, A.Z., O.I., A.E.A., B.W. and M.D.-M.; investigation, A.Z., O.I., A.E.A., B.W., M.D.-M., G.W. and R.S.; resources, G.W., A.E.A. and M.D.-M.; data curation, A.Z., O.I., A.E.A., B.W. and M.D.-M.; writing—original draft preparation, A.Z., O.I., A.E.A., B.W. and M.D.-M.; writing—review and editing, R.S., G.W., M.T.R. and K.D.; visualization, M.D.-M.; supervision, G.W.; project administration, A.Z. All authors have read and agreed to the published version of the manuscript.

**Funding:** This research received no external funding.

**Institutional Review Board Statement:** Not applicable.

**Informed Consent Statement:** Not applicable.

**Data Availability Statement:** No new data were created or analyzed in this study. Data sharing is not applicable to this article.

**Conflicts of Interest:** The authors declare no conflicts of interest.

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
