# Peer review of "Barriers and Challenges Affecting Quality Education (Sustainable Development Goal #4) in Sub-Saharan Africa by 2030"

_sustainability, doi:10.3390/su16072657_

Round 1
Reviewer 1 Report
Comments and Suggestions for Authors
I fully agree with one of the statements of the paper. "Inadequate curriculum may also be mitigated through culturally relevant curriculum. Initially, ensuring that curriculum is delivered in students’ home language could increase student retention and understanding of content."
I think that the staudy could be an important contribution to the relevant authorities ands stakeholders in their activities to improve the education system in SSA countries.
Reviewer 2 Report
Comments and Suggestions for Authors
Thank you for the opportunity to review this paper- it is an interesting and important work. My main comments relate to methods. Was this a narrative or systematic review? There is not enough detail in the methods for a systematic review but more detail than is seen in some narrative reviews. Please clarify and, if a systematic review, include further detail as required by checklists such as PRISMA.
Comments on the Quality of English LanguageSome parts could be reworded for succinctness and clarity. For example, I am not sure what you are trying to say in the first few sentences of section 5. Conclusions.
Reviewer 3 Report
Comments and Suggestions for Authors
The paper, Review of Barriers and Challenges Affecting Quality Education (SDG#4) in sub-Saharan Africa by 2030, is very well-written and interesting. The authors discuss the problems limiting education is SSA and provide some general suggestions on how to resolve those problems. My one criticism is that, for the most part, the suggested improvements are very general. It would be nice to see some specific, concrete recommendations for remediating the educational barriers facing students, particularly for girls and for special needs students.
In the article, Review of Barriers and Challenges Affecting Quality Education (SDG #4) in sub-Saharan Africa by 2030, the authors discuss problems that are facing the educational systems in sub-Saharan Africa and how those challenges are likely t extend into the next decade and beyond. The authors also discuss some possible ways of addressing the challenges. Educational challenges are a worldwide phenomenon although the specifics may be different for different regions. Understanding the challenges facing a region is the first step towards addressing those challenges and the authors do a good job outlining these challenges and discussing how they are impacting, and will continue to impact, sub-Saharan Africa.
The manuscript provides information about issues such as the lack of school funding and teacher education that, while present in other countries, do not take on the same form. The authors do a good job describing what this looks like in sub-Saharan Africa. They address issues, such as gender inequality in education, that may not be a prevalent issue in other countries and discuss how these discrepancies impact both the students and the region in the short and long term.
A quick perusal of the literature showed that the most recent academic papers on the topic are from 2007 and 2010. The current manuscript draws on the most recent data and therefore helps to update knowledge about and understanding of the challenges facing education in an ever-changing landscape. The one weakness of the paper is that the suggestions for improvements are very general. I would improve the manuscript if the authors included some concrete recommendations for remediating the challenges facing students, particularly for girls and special needs students.

Reviewer 4 Report
Comments and Suggestions for Authors
Dear authors,
the content of your article impressed me. The information reflecting the situation in a number of African countries is very relevant and meets the requirements of the journal, revealing the significance of the main goals of the UN - quality of education, gender equality. The author's reference to contemporary sources and articles is noteworthy, which makes the research findings unquestionable and reliable. Although this article does not present the results of an empirical study, the literature review is quite convincing. As a recommendation, we would like to draw attention to the key words, which can be slightly specified in accordance with the research material when applying content analysis. The paper would have been significantly benefited if the authors had emphasised their personal contribution to the problem in the discussion section. The logic chain to support this research problem may be not clear. Stating the hypotheses of the study would allow the authors' research to be structured. And in the conclusion section, further research perspectives and limitations of the theoretical study should be added. In my opinion, the facts reflected should be printed with minor improvements at the editors' discretion. I would recommend future empirical studies for the authors to have solid foundations that make their research results more applicable.
Round 2
Reviewer 2 Report
Comments and Suggestions for Authors
Thabk your for addresing the reviewer's comments so quickly. I have reviewed the revised version of the paper and deem it suitable for publication.
Author Response
We thank the reviewer for an expedited review of the revised manuscript. We look forward to others' reviews in the coming days.
The Authors